# Carrageenans and the Carrageenan-Echinochrome Complex as Anti-SARS-CoV-2 Agents

**DOI:** 10.3390/ijms26136175

**Published:** 2025-06-26

**Authors:** Natalya V. Krylova, Anna O. Kravchenko, Galina N. Likhatskaya, Olga V. Iunikhina, Valery P. Glazunov, Tatyana S. Zaporozhets, Mikhail Y. Shchelkanov, Irina M. Yermak

**Affiliations:** 1G.P. Somov Institute of Epidemiology and Microbiology, Rospotrebnadzor, 690087 Vladivostok, Russia; olga_iun@inbox.ru (O.V.I.); niiem_vl@mail.ru (T.S.Z.); adorob@mail.ru (M.Y.S.); 2G.B. Elyakov Pacific Institute of Bioorganic Chemistry, Far-Eastern Branch of the Russian Academy of Science, 690022 Vladivostok, Russia; kravchenko_89@mail.ru (A.O.K.); galin56@mail.ru (G.N.L.); glazunov@piboc.dvo.ru (V.P.G.)

**Keywords:** carrageenans, SARS-CoV-2, molecular docking, antiviral activity

## Abstract

The diversity of structural types of carrageenans (CRGs)—sulfated polysaccharides of red algae—determines their different biological activities. The different types of CRGs (kappa, lambda, kappa/beta-CRGs) were isolated from the red algae of the Pacific coast. Molecular docking was performed to determine potential interactions of CRGs with the receptor-binding domain (RBD) of SARS-CoV-2 and its cellular receptor—angiotensin—converting enzyme type 2 (ACE2). CRGs interacted with ACE2 and RBD via hydrogen bonding and ionic interactions. The strongest binding affinity of CRGs and ACE2 was observed for kappa-CRG. Molecular docking was confirmed by results studying the effects of CRGs against SARS-CoV-2 in vitro. The ability of CRGs, as well as the complex CRG with sea urchin echinochrome (Ech), to inhibit SARS-CoV-2 replication in Vero E6 cells was studied using cytopathic effect (CPE) inhibition and RT-PCR assays. The simultaneous treatment of cells with CRGs and the virus revealed that kappa-CRG exhibited the most significant antiviral effect among all the polysaccharides, with a selective index (SI) of 33. The kappa-CRG/Ech complex exhibited the highest virucidal effect on SARS-CoV-2 particles with an SI above 70 (more than two times higher than that of CRG and Ech) and reduced viral RNA levels by 45% (IC = 45%). Our results illustrate that CRGs and kappa-CRG/Ech complex can act as protective agents against SARS-CoV-2.

## 1. Introduction

Biologically active substances from natural sources are characterized by structural diversity and exhibit a wide range of physiological effects. Recently, unique natural compounds from various marine sources have been investigated as potential antiviral agents against SARS-CoV-2 [1]. The global COVID-19 pandemic, caused by the rapid spread of the SARS-CoV-2 virus, resulted in over 750 million confirmed cases and 6.9 million deaths worldwide between 11 March 2020, and 5 May 2023 [2,3]. The World Health Organization has announced that COVID-19 is no longer a global health emergency and is now considered an “entrenched and persistent problem” [4]. However, despite significant advances in genome mapping, diagnostics, and vaccine development, which have helped control the pandemic and reduce mortality rates, SARS-CoV-2 continues to circulate in all countries. The emergence of new virus genotypes with unpredictable genetic variability and antigenic properties underscores the ongoing need for the research and development of effective anti-SARS-CoV-2 agents.

In this regard, the study of natural non-toxic and biocompatible compounds that prevent the adhesion of viruses to host cells and inhibit their replication is of particular importance. Currently, much attention has been given to the potential of seaweed sulfated polysaccharides (PSs), which are unique in their structure and properties as antiviral agents for the prevention of COVID-19. It has been shown that PSs possess high antiviral activity, preventing the penetration of DNA and RNA viruses into host cells through interactions with positively charged glycoproteins of the viral envelope [5,6,7]. Most seaweed-derived sulfated PSs have been reported to effectively inhibit SARS-CoV-2 entry. The polyanionic nature of PSs is a crucial factor, and the antiviral activity depends quantitatively and qualitatively on their structural architecture [7,8,9]. The key factors influencing the antiviral activities of PSs include the degree of sulfation, the specific position of sulfate groups, molecular weight, and total PS content in the inhibitory effect against SARS-CoV-2 host cell entry [5,6].

A special place among seaweed sulfated PSs is occupied by carrageenans (CRGs), which have no structural analogs among other plant PSs and have demonstrated activity against a variety of viruses, including SARS-CoV-2 [7,8,9]. CRGs are linear sulfated galactans, whose basic structural units are composed of disaccharide carrabiose, consisting of alternating β-1,3- and α-1,4-linked galactose residues. Variations in the basic structure are determined by the content of 3,6-anhydrogalactos, and the number and position of sulfate groups. Since natural CRGs are mixtures of non-homologous PSs, the term «disaccharide repeating unit» refers to an idealized structure [10]. The three most industrially exploited types of CRGs, namely k-, i-, and λ-CRGs, are distinguished by the presence of one, two, and three ester sulfate groups per repeating disaccharide unit, respectively. Units of unsulfated carrageenose are designated as (β-CRG). Natural CRGs usually contain repeating disaccharide units of several types, forming hybrid structures with a predominance of one type such as κ/β-hybrids, κ/i-hybrids, κ/μ-hybrids, or v/i-hybrids [11,12]. The structural features of CRGs depend on the type of algae and its life cycle phase. Depending on their structure, CRGs exhibit varying antiviral activities [8,12,13]. Previously, we demonstrated that CRGs isolated from red algae of the Far Eastern seas inhibited the replication of herpes simplex virus type 1 (HSV-1) and enterovirus B (ECHO-1) and that their antiviral activity depended on the structure of the CRGs [13].

The in vitro antiviral properties of lambda- and iota-CRGs-rich PS fractions isolated from *H. floresii* and *S. chordalis*, respectively, against the Wuhan type of SARS-CoV-2 were presented by Jousselin et al. [14]. The effect of CRGs on viral replication was assessed during infection of human airway epithelial cells with a clinical strain of SARS-CoV-2. The addition of CRGs at different intervals during the infection helped to determine their mechanism of antiviral action. The authors suggest that the antiviral effect of CRGs is due to the inhibition of viral attachment to the cell surface and suggest that these PSs could be used as a first-line treatment in the respiratory tract to inhibit SARS-CoV-2 infection and transmission [14]. Jang et al. hypothesized that lambda-CRGs, which were effective against SARS-CoV-2, interfere with the virus’s attachment to cell surface receptors and subsequently prevent virus entry. Moreover, these authors experimentally demonstrated that commercial lambda-CRGs used as nasal drops in mice reduced weight loss caused by influenza infection and prevented infection-related death [15]. Nasal sprays may be an effective option for combating SARS-CoV-2 infection, as the nasal cavity and nasopharynx are the initial sites of virus replication. Several nasal sprays containing CRGs have been tested against respiratory viruses. Schütz et al. showed that both nasal and oral sprays containing kappa-CRG inhibited SARS-CoV-2 replication in human airway epithelial cells [16]. A potent antiviral effect of iota-CRG in MucilAir nasal cultures was observed when applied before and repeatedly after SARS-CoV-2 infection. It was shown that topically applied iota-CRG could effectively prevent SARS-CoV-2 infection and replication [17]. The results of the study by Shruti Bansal and co-authors showed that iota-CRG significantly inhibited SARS-CoV-2 in vitro opening the prospect for clinical use of this nasal spray for the prevention and early treatment of COVID-19 [18]. Moreover, the authors found that combining iota-CRG with xylitol could be a promising strategy for nasal spray formulation. Of particular note is a pilot study of an iota-CRG nasal spray used alongside standard preventive measures, which showed a significant reduction in the risk of SARS-CoV-2 transmission to healthy patients undergoing treatment as well as hospital staff [19].

It is known that the penetration of SARS-CoV-2 into target cells begins with the specific interaction of the receptor-binding domain (RBD) of the S1 subunit of the spike (S) protein of the virus with the peptidase domain (PD) of the cellular receptor—angiotensin-converting enzyme type 2 (ACE2) [20,21]. The SARS-CoV-2 spike glycoprotein and human angiotensin converting enzyme 2 are two key targets for the prevention and treatment of COVID-19. One strategy for antiviral therapy may involve chemical compounds that interfere with or destabilize the ACE2-RBD complex in its pre-fusion state, thereby preventing viral infection at an early stage.

To investigate the structure-activity relationship, a series of PS from *Saccharina japonica* were investigated for their binding abilities to pseudotype particles containing the SARS-CoV-2 SGP and ACE2 using surface plasmon resonance. The authors showed that sulfated galactofucan and glucuronomannan strongly inhibited the interactions between SARS-CoV-2 SGP and heparin, whereas inhibition of the interaction between SARS-CoV-2 SGP and ACE2 was very weak [22].

It is known that during viral infection, individuals face an increased risk of pathogenicity caused by free radicals. In this context, antioxidants play a decisive role in viral pathogenesis [23]. Echinochrome (7-ethyl-2,3,5,6,8-pentahydroxy-1,4-naphthoquinone), a quinoid pigment of sea urchin, has pronounced antioxidant activity [24]. Over the past decade, the mechanisms underlying its antioxidant [25] and antiviral [26,27] properties have been studied in detail. We previously showed that CRGs improve the solubility of water-insoluble Ech and prevent its oxidative degradation [28]. Additionally, we demonstrated that CRGs interact with Ech to form a complex with greater antiherpetic activity than their constituent compounds [29].

This study aimed to assess the anti-SARS-CoV-2 activity of different types of CRGs (kappa-, lambda-, and kappa/beta-CRGs) isolated from red algae of the Far Eastern seas as well as the kappa-CRG complex with echinochrome and in silico assessment of potential interactions of CRGs with both the RBD of SARS-CoV-2′s S-protein and human ACE2.

## 2. Results

### 2.1. Characteristics of CRGs

The polysaccharide was extracted from seaweed *Chondrus armatus* and *Tichocarpus crinitus* purified from impurities and separated using 4% KCl into insoluble (gelling) and soluble (non-gelling) fractions, as described in the Section 4 [30]. Gelling fractions (KCI-insoluble) from both algae species and one non-gelling fraction from *C. armatus* were characterized and used in the work. The polysaccharide isolated from *C. armatus* and purified by ultrafiltration from low molecular weight impurities, designated as (Σ-CRG) and used in further work. According to chemical analysis all polysaccharides contain only carrabiose units consisting of sulfated galactose and 3,6-anhydro-drogalactose.

To clarify the structure of the fractions isolated polysaccharides, Fourier transform infrared (FTIR) and NMR spectroscopies methods were used. Absorption bands in the IR spectra and chemical shifts in the NMR spectra were assigned via comparison to signals of known carrageenan structures [10,11,31,32]. The obtained spectra also were compared with spectra of polysaccharides isolated by us from these species of algae, as detailed previously [30,33,34,35]. Based on ^13^C-NMR and FTIR spectra, all polysaccharides were identified as carrageenans.

In the FTIR spectra of all fractions, polysaccharides showed a broad and strong absorption band with a maximum at about 1225–1240 cm^−1^ indicating the presence of sulfate groups (–S=O asymmetric vibration), which is in agreement with results of chemical analysis (Figure 1). The infrared spectra KCI-insoluble fractions polysaccharides from *C. armatus* and *T. crinitus* showed absorption bands at 932 cm^−1^ for 3,6-anhydrogalactose (C(3′)–O–C(6′) stretching vibration) and the stretching vibration S–O(4) bonds of 848 cm^−1^, which is characteristic of the axial sulfate group at C-4 of the 3-linked β-D-galactose. This made it possible to assign the polysaccharides to kappa-type of CRG. Moreover, the absorption band at 892 cm^−1^ in the IR spectrum of the insoluble fractions of *T. crinitus* also evidenced the presence of non-sulfated β-D-galactose residues, which is typical for β-CRG. Thus, FTIR spectroscopy data suggest that KCl-insoluble polysaccharide from *C. armatus* was represented by kappa-CRG [30], whereas KCl-insoluble polysaccharide fraction from *T. crinitus* had hybrid structures and was identified as kappa/beta-CRG [33]. There was no absorption band corresponding to 3,6-anhydrogalactose in the IR spectra of the soluble fraction of *C. armatus* consistent with chemical analysis. At the same time, in the FTIR spectra of this fraction, a broad absorption band was observed at 830 cm^−1^, and a shoulder at about 820 cm^−1^, which corresponds to the primary equatorial sulfate group at C-6 and the secondary equatorial sulfate group at C-2 of 4-linked α-D-galactose, characteristic of lambda-carrageenan [30]. According to a comparative analysis of spectral data and the identity of the spectra, the KCI-insoluble fraction from *C. armatus* was kappa-CRG; KCl-insoluble polysaccharides from *T. crinitus* had hybrid structures and have been identified as kappa/beta-CRG with the ratio of kappa-units to beta: 60:40 [33]. The KCI-soluble fraction *C. armatus* corresponded to lambda-type CRG. So, the unfractionated Σ-CRG consists of kappa (κ) and lambda (λ)-CRGs type with the ratio 60:40 [30,34].

FTIR spectroscopy data were confirmed by NMR spectroscopy. The ^13^C-NMR spectrum of KCI-insoluble fraction polysaccharides from *C. armatus* showed two signals in the anomeric carbon resonance area. The signal at 103.2 ppm was characteristic of the C-1 of 1,3-linked β-D-galactose residue (G4S) and the signal at 95.8 was assigned to C-1 of 1,4-linked 3,6-anhydro-α-D-galactose (DA) of κ-carrageenan. The signal C-4 of 1,3-linked β-D-galactose in the ^13^C-NMR spectrum was shifted to 8 ppm in the weak field, which indicated the presence of a sulfate group at the position. The ^13^C-NMR spectrum in the upfield region was typical for κ-carrageenan (Figure 2A).

In contrast to this spectrum of KCI-insoluble fraction polysaccharides from *C. armatus*, the ^13^C-NMR spectrum of the KCI- insoluble fraction of the polysaccharide from *T. crinitus* contained two poorly resolved signals at 103.1 and 102.7 ppm in the anomeric carbon resonance area. This was due to the overlap of the C-1 signals of 3-linked β-D-galactose 4-sulfate κ-CRG (G4S) and 3-linked β-D-galactose (G) β-CRG. Here were two signals observed at 95.1 ppm and at 94.4 ppm, which were characteristic of C-1 the 4-linked 3,6-anhydro-α-D-galactose κ-CRG (DA) and β-CRG (DA’), respectively (Figure 2B).

The chemical structures of the disaccharide repeats units of the three types of CRGs are presented in Table 1. These types of CRGs differ by number and position of the sulphated groups, and by the presence (κ) or absence (λ) of a 3,6-anhydrogalactose unit. The degree of sulfation decreases in the following row: λ > κ > κ/β. β-CRG does not contain sulfate groups.

### 2.2. Molecular Docking of Carrageenan Tetrasaccharides with SARS-CoV-2 RBD and ACE2

The method of molecular docking and dynamic modeling has been applied in recent years to assess the binding affinity between two macromolecular targets. The human angiotensin-converting enzyme 2 (hACE2) receptor is known, and it is recognized by the spike protein of SARS-CoV-2 for initiating infection [21]. To identify potential binding sites, a known sulfated polysaccharide (e.g., heparin) was used as a molecular probe, which has been shown to exhibit significant antiviral activity against SARS-CoV-2 by binding to the RBD, preventing the virus from targeting ACE2 [36,37].

We used molecular docking to ascertain potential interactions CRG with both the receptor-binding domain (RBD) of SARS-CoV-2′s spike protein and human angiotensin-converting enzyme-2 (ACE2). First of all, to determine the molecular docking site of CRG with RBD and hACE2, it was important to find the binding site of SARS-CoV-2 Delta RBD and hACE2 [38], which we assessed using the MOE program. The RBD and ACE2 binding site was determined in the structure of the complex (PDB ID 7w9i) using the SiteFinder module of the MOE program and is reproducible (Figure 3). Binding site amino acid residues were used for docking, as with RBD and with hACE2.

Contact analysis revealed the presence of positively charged lysine residues at the site, which could be a potential binding site for the negatively charged CRG. Next, we studied in silico the possible interaction of CRG with RBD and ACE2. For this purpose, we used three types of carrageenan (kappa-, lambda-, and beta- units) whose structures are presented in Table 1. According to the chemical structure, kappa-CRG contains one sulfate group, lambda three sulfate groups and the disaccharide unit b-CRG does not contain sulfates.

For convenience of calculation in molecular docking, we used tetrasaccharides of these types of CRG obtained using the molecular editor of the MOE program. Molecular docking of these CRGs with the ACE2 binding site was carried out (Figure 4). Molecular docking has shown that the docking scores for CRG varies from −7.7 to −6.7 kcal/mol and is ranked in the order κ-CRG > β-CRG = λ-CRG.

Analysis of contacts between CRGs and ACE2 showed the involvement of lysine residues in the binding of all types of CRGs (Figure 4; Appendix A). The contacts of CRGs tetrasaccharides with the human ACE2 domain are presented in Appendix A. According to the calculated data given in Appendix A, the interaction of CRGs with ACE2 occurs due to the formation of H-donor, H-acceptor, and ionic bonds. As can be seen, the highest binding energy of CRGs and ACE2 was observed for kappa-CRG (Appendix A).

At the next stage, CRG was docked with RBD. As in the case of ACE 2, for all types of CRGs, binding contacts with RBD were observed. Molecular docking of CRGs with the RBD binding site was carried out (Figure 5). For RBD, the docking score for CRG ranged from −5.8 to −5.2 kcal/mol. Analysis of contacts between CRGs and RBD showed the involvement of arginine residues in the binding of all types of CRGs. As in the case of ACE 2, the interaction of CRGs tetrasaccharides with RBD occurs due to the formation of H-donor, H-acceptor, and ionic bonds, the energy calculation of which is presented in Appendix A. The largest number of ionic bonds was observed for λ-CRG (Appendix A).

### 2.3. Cytotoxicity and Anti-SARS-CoV-2 Activity of the Different Types of Carrageenans

A study of the cytotoxicity of the different samples of carrageenans (CRGs) from *C. armatus* (kappa-CRG, lambda-CRG, Σ-CRG) and kappa/beta-CRG from *T. crinitus* and reference compounds (ribavirin and remdesivir) against Vero E6 cells was carried out using the methyl-thiazolyltetrazolium bromide (MTT) test. Cytotoxicity for each tested compound was expressed as the 50% cytotoxic concentration (CC_50_), which reduced the viability of treated cells by 50% compared to untreated cells, and the maximum non-toxic concentration (MNTC), at which cell survival exceeded 90%. The investigated carrageenans had low cytotoxicity: their CC_50_ was above 2000 μg/mL, MNTC—250 μg/mL. At the same time, CC_50_ and MNTC of ribavirin were 730 μg/mL and 150 μg/mL, remdesivir—72 μg/mL and 12.5 μg/mL, respectively (Appendix A). To further study the anti-SARS-CoV-2 activity of the tested compounds, concentrations of carrageenans and reference compounds within or below MNTC were used.

The inhibitory effects of the different types of CRGs on SARS-CoV-2 infection process in Vero E6 cells were evaluated by the cytopathic effect (CPE) inhibition assay using MTT-test and virus replication level (VRL) inhibition assay using real-time RT-PCR assay. To select the CRG that most actively inhibits the cytopathogenic effect of the virus and reduces the level of SARS-CoV-2 RNA, cells were simultaneously treated with the virus and the tested carrageenans, which allow us to evaluate the effect of polysaccharides on both cells and the virus. For CPE inhibition assay, the results of the virus-inhibitory activity of tested compounds were used for calculations of the 50% inhibitory concentration (IC_50_) and the selectivity index (SI) for each of the compounds. For RT-PCR assay, effect tested compounds on viral load was assessed using the inhibition coefficient (IC, %).

It was found that the all investigated carrageenans had antiviral activity, which was confirmed by the inhibition of the cytopathic effects induced by SARS-CoV-2 (Figure 6 and Appendix A). The kappa-CRG demonstrated the greatest antiviral effect: its IC_50_ = 61 µg/mL and SI, indicating the efficacy and safety of the compound, was 33, while the values of these parameters for less sulfated kappa/beta-CRG were 2.5 times lower (IC_50_ = 160 µg/mL and SI 12.5) than for kappa-CRG. Meanwhile, ribavirin showed the lowest anti-SARS-CoV-2 activity (IC_50_ = 207 µg/mL, SI 3.5), and remdesivir—the highest (IC_50_ = 1.4 µg/mL, SI 51).

Using RT-PCR assay, it was shown that different types of CRGs at MNTC (250 μg/mL) significantly inhibited SARS-CoV-2 replication (Appendix A). Among the investigated carrageenans, kappa-CRG caused the highest reduction in viral RNA levels with IC 36%. Herewith, remdesivir, which is used as a positive control for viral inhibition [39], at MNTC (12.5 μg/mL) showed SARS-CoV-2 inhibition with IC 49% (Appendix A).

### 2.4. Anti-SARS-CoV-2 Activity of the κ-CRG/Ech Complex and Its Components

Based on the most active kappa-CRG from *C. armnatus*, its complex with the antioxidant Ech—κ-CRG/Ech complex—was developed. The effect of the obtaining complex and its components on the early stages of the SARS-CoV-2 life cycle was studied using cytopathic effect (CPE) inhibition assay and real-time RT-PCR assay. For that purpose, cells were pre-incubated with the κ-CRG/Ech complex and its components before the viral challenge (pretreatment of cells); virus was pre-incubated with the investigated compounds before infection (pretreatment of virus); and compounds were added to cells during the adsorption of SARS-CoV-2 (simultaneous treatment) and after adsorption and penetration of virus to the cells (treatment of infected cells).

The study of the antiviral activity of the investigated compounds using a CPE inhibition assay revealed that the κ-CRG/Ech complex most effectively suppressed the replication of SARS-CoV-2 when the virus was pretreated with the complex (direct virucidal effect). At the same time, κ-CRG/Ech exhibited significantly higher antiviral activity compared to kappa-CRG and Ech: the SI of the complex (SI 77) was more than two times higher than that of κ-CRG and Ech (*p* ≤ 0.05) (Table 2). Ribavirin and remdesivir, with this method of compound application, did not show antiviral activity. However, pretreatment Vero E6 cells with compounds before infection (preventive effect) showed that κ-CRG/Ech complex protected cells against coronavirus infection with an SI 3.5 times lower than the SI of the κ-CRG (*p* ≤ 0.05), which can be explained by the cytotoxicity of Ech. Ribavirin and remdesivir did not show antiviral activity. Significant inhibition of SARS-CoV-2 replication was observed after simultaneous treatment of cells with the virus and the studied compounds. The CRG/Ech complex showed higher inhibitory activity (SI 42) than kappa-CRG (SI 32) (*p* ≤ 0.05) and ribavirin (SI 3.5) but lower than remdesivir (SI 51). The application of κ-CRG/Ech complex and κ-CRG after virus adsorption and penetration to cells (treatment of infected cells) had a moderate effect on SARS-CoV-2 replication (the average SI was 22).

The results of the study of the effect of the κ-CRG/Ech complex and its components on the early stages of the SARS-CoV-2 life cycle using RT-PCR assay are presented in Figure 7 and in Appendix A. A comparative analysis of the anti-SARS-CoV-2 activity of the investigated compounds showed that with direct effect on the virus and with simultaneous treatment of cells with the virus and the compound, κ-CRG/Ech complex inhibited viral replication more effectively than its components (*p* ≤ 0.05). At the same time, pre-treatment of cells with the compounds before infection revealed that κ-CRG caused a higher reduction in viral RNA levels compared to the κ-CRG/Ech complex (*p* ≤ 0.05). Meanwhile, the application of κ-CRG/Ech and kappa-CRG after virus penetration to cells (1 h post-infection) showed moderate inhibition of viral replication (average IC values 18%), in contrast to the reference medicine remdesivir (IC 56.3%) (Figure 7 and Appendix A).

Thus, the CPE inhibition and RT-PCR assays showed that the tested carrageenans, especially kappa-CRG from *C. armnatus*, possessed significant anti-SARS-CoV-2 activity when added simultaneously with the initiation of viral infection. The κ-CRG/Ech complex, in turn, demonstrated high antiviral activity, mainly due to its direct virucidal effect.

## 3. Discussion

Research into the development of drugs for the prevention and treatment of SARS-CoV-2 infection continues to be pertinent. Given that the hACE2 receptor, targeted by the S protein of SARS-CoV-2 to initiate infection, is highly expressed in the nasal and oral mucosa [40], intranasal or intraoral delivery of antiviral agents could be a valuable strategy in stopping the spread of the virus. Red algal PS, particularly CRGs, has shown effectiveness against various viruses and has been extensively researched in clinical trials for viral diseases [15,19]. Recently, a successful antiviral effect of a CRG-based nasal spray was demonstrated against rhinoviruses and SARS-CoV-2 [16].

The most commonly employed approach in the quest for new drugs is a method of molecular docking and dynamic modeling, which has been used widely in recent years to assess the binding affinity between two macromolecular targets. Molecular docking analyzes the conformation of molecules within the binding site of a macro-molecular target [41]. Experimental molecular docking data have been reported in the literature to evaluate the interaction of sulfated PS with both the SARS-CoV-2 S protein RBD and human ACE2. Modeling has shown that the sulfated PS heparin exhibits antiviral activity against SARS-CoV-2 by binding to the S protein; thereby inhibiting the virus’s attachment to ACE2 and its penetration into the host cell [37]. Salih and co-authors assessed the antiviral potential of PSs by analyzing their interactions with various targets using several in silico tools. This approach enabled the authors to identify the most promising candidates against SARS-CoV-2 [36].

In the present study, molecular docking was used to study the interaction of CRGs with ACE2 and RBD and to evaluate potential binding sites. For modeling, we used the tetrasaccharides of the corresponding type CRGs: kappa-, beta-, and lambda-CRGs as a molecular probe that can prevent the virus from targeting ACE2. Their disaccharide units differ in both degree of sulfation and the presence of 3,6-anhydrogalactose units.

Molecular docking revealed that CRG types were able to bind to both ACE2 and RBD. The interactions between CRGs and RBD involved arginine residues in the binding of all types of CRGs. The largest number of ionic bonds was observed in lambda-CRG, which corresponds to its high degree of sulfation. Molecular docking of CRGs to the ACE2 binding site showed CRGs demonstrated a higher binding affinity for the ACE2 protein target than for RBD.

As in the case of RBD, the interactions between CRG and ACE2 occurred due to the formation of H-donor, H-acceptor, and ionic bonds. The highest binding energy between CRGs and ACE2 was observed for kappa-CRG. Our data are consistent with recent molecular docking studies [42]. In this article, ACE2 was used as a molecular target for screening kappa-, lambda-, and iota-CRGs and it was shown that iota-CRG had the greatest binding affinity. Iota-CRG, like kappa-CRG, contains the sulfate groups (two and one, respectively) and 3,6-anhydrogalactose in contrast to lambda-CRG, which does not contain this monosaccharide residue. Therefore, the presence of 3.6-anhydrogalactose is important for enhancing CRG affinity for ACE2.

The molecular docking results on CRG interaction with ACE2 were confirmed in vitro. We investigated the anti-SARS-CoV-2 activity of kappa-CRG, kappa/beta-CRG, and Σ-CRG (kappa+lambda), which all contain kappa-tetrasaccharide units, showing the highest ACE2 binding energy. Additionally, we tested lambda-CRG, which, according to modeling, had the largest number of ionic bonds with RBD.

The antiviral activity of CRGs was tested on Vero E6 cells against SARS-CoV-2 Delta Variant (GenBank ID ON692745.1). The effect of CRGs during the early stages of the SARS-CoV-2 life cycle was investigated using CPE inhibition and RT-PCR assays. Simultaneous treatment of Vero E6 cells with CRGs and SARS-CoV-2 virus showed that all CRG samples inhibited virus-induced cytopathic effects and reduced viral RNA levels. As shown in Figure 6 and Appendix A, kappa-CRG showed the greatest activity: its 50% inhibitory concentration (IC_50_) was 61 μg/mL, and its selectivity index was 33; it inhibited SARS-CoV-2 replication with an inhibition coefficient (IC) of 36%. This is in good agreement with the results of molecular docking, which showed the highest binding energy kappa-CRG with the cellular receptor ACE2. Typically, the antiviral activity of PSs correlates with their molecular weight (Mw). It is known that CRGs with relatively high molecular weight can significantly suppress viral replication by inhibiting virus attachment to the host cell [43]. All CRGs used in this study had high Mw (>100 kDa) with kappa-CRG showing the greatest antiviral activity having the highest Mw. Previously, we also demonstrated, that both the parent κ/β-CRG and its oligosaccharide exhibited potent antiviral activity at non-toxic doses as potential HIV-1 inhibitors [44,45].

The high activity of kappa-CRG may be attributed to the unique chemical structure of this PS, which has a high molecular weight and also contains 3.6-anhydrogalactose in the polymer chain. This residue adopts a 1C4 chair conformation resulting in the formation of a helical structure [46] that may form a mesh on the cell surface, thereby preventing viral attachment and inhibiting infection efficiency.

Among the PS samples studied, the lowest antiviral activity was observed for the less sulfated kappa/beta-CRG. Although both kappa-CRG and kappa/beta-CRG share a common monosaccharide composition containing galactose and 3.6-anhydrogalactose—the β-unit in kappa/beta-CRG does not contain sulfates. The lower degree of sulfation likely explains the weaker activity of kappa/beta-CRG compared to kappa-CRG, consistent with molecular docking data showing lower binding energy to ACE2.

Thus, our molecular docking results confirmed the in vitro data and provided insights into the mechanism of the anti-SARS-CoV-2 activity of CRGs. CRGs can directly bind to both the RBD of the SARS-CoV-2 S glycoprotein and its cellular receptor ACE2, blocking their interaction and preventing viral attachment and entry into host cells. Among the PSs studied, kappa-CRG demonstrated the greatest affinity corresponding to its highest binding energy with ACE2.

In this study, we used kappa-CRG as a mucoadhesive matrix to include the red pigment of sea urchins, echinochrome (Ech). Our previous results showed that CRGs improve Ech solubility, protect it from oxidation, and form a stable complex with enhanced antiviral activity against herpes [28,29]. We also previously demonstrated via molecular docking that Ech interacts with CRG to form a stable complex [47]. Such a complex, in which the water-insoluble antioxidant Ech is protected from oxidation, may serve as a novel antiviral agent, considering the anti-SARS-CoV-2 activity of CRGs demonstrated in this study.

We investigated the anti-SARS-CoV-2 activity of the resulting κ-CRG/Ech complex and its components using CPE inhibition and RT-PCR assays. To explore the mechanism of action at the early stages of virus–cell interaction, several additional treatment schemes involving both the virus and Vero E6 cells were employed. It was found that κ-CRG/Ech most effectively suppressed the replication of SARS-CoV-2 when the complex was directly exposed to viral particles (direct virucidal effect): achieving an SI of 77 and reducing viral load with an inhibition coefficient (IC) of 45%. Moreover, the antiviral activity of the κ-CRG/Ech complex was more than two times higher than that of kappa-CRG and Ech (*p* ≤ 0.05). The complex also effectively inhibited virus–cell interactions when added simultaneously with the initiation of viral infection. Thus, our investigation of the complex during early SARS-CoV-2 infection stages revealed that its primary antiviral mechanism is direct interaction with the virus (virucidal activity). A comparative analysis of the anti-SARS-CoV-2 activity of the κ-CRG/Ech complex versus its components demonstrated that complex more effectively inhibited viral replication under both direct and simultaneous treatment conditions (*p* ≤ 0.05). We have previously shown on the herpes virus that Ech exhibits a virucidal effect, while CRG mainly blocks the attachment of the virus to the cell [29]. In the present work, their synergistic combined effect is also manifested in an increase in the antiviral activity of the complex against SARS-Co-2.

Our data demonstrate that the structural features of CRGs influence their ability to inhibit SARS-CoV-2 replication. Incorporating Ech into a kappa-CRG enhances antiviral effects, suggesting that CRG/Ech complex may be useful for oral, nasal, or buccal applications during infection.

The in silico results allowed us to better understand the complex mechanism underlying the action of CRGs and its complex against SARS-CoV-2 in vitro. As is known, SARS-CoV-2 uses its receptor-binding domain RBD (Spike glycoprotein) to interact and gain access to host cells by binding to the ACE2 receptor [20,21]. The S protein-ACE2 interaction is the primary key to virus entry. In our case, molecular docking clearly showed that CRGs bind to both RBD and ACE2 and, thus, block their interaction and the entry of the virus into host cells. At the same time, CRGs exhibit a high affinity for ACE2 and, thus, already at the initial stage of infection, compete with RBD and suppress the attachment of the virus to the cell. Thus, we believe that different mechanisms are involved in the action of the studied polysaccharides again SARS-CoV-2, the combination of which increases the effectiveness of the drugs.

Currently, new variants of SARS-CoV-2 are circulating, many of which display more electropositive regions on the RBD surface at the ACE2 binding site [48,49]. This suggests that the virus-inhibitory activity of negatively charged CRGs of various types will remain effective in targeting the binding of new SARS-CoV-2 variants to ACE2. Since additional mutations in the S protein outside the RBD, as well as other co-receptors and cofactors, may influence infection with new variants our findings on the antiviral activity of κ-CRG and its Ech complex support further investigation of these compounds as potential antiviral agents against SARS-CoV-2.

## 4. Materials and Methods

### 4.1. Algal Material

Red alga *Chondrus armatus* (Gigartinales: Gigartinaceae) and *Tichocarpus crinitus* (Gigartinales: Tichocarpaceae) were harvested at Peter the Great Bay, Sea of Japan, and identified based on morphological and anatomical characteristics by Prof. E. Titlyanov and T. Titlyanova (National Scientific Center of Marine Biology, Far-Eastern Branch of the Russian Academy of Sciences) using a transmission electron microscope. According to the identification, the selected seaweed was in the vegetative form, lacking any reproductive organs. The algae were washed with tap water to remove excess salt. Bleaching of the seaweed was performed by maintaining the specimen in pure acetone for 3 days prior to being dried in the air.

### 4.2. Extraction and Characterization of Carrageenans

Dried and milled algae (50 g) were suspended in hot water (1.5 L) and the polysaccharides were extracted three times at 80 °C for 3 h in boiling water. Hot extracts were combined, centrifuged at 4000 rpm^−1^ to remove residues of the cell wall, filtered through a Vivaflow200 membrane (Sartorius, Germany) with a pore size of 100 kDa, concentrated on a rotary evaporator, and precipitated polysaccharides with a triple volume of 96% ethanol. The precipitates as the crude extracts were purified by redissolving in water and were concentrated, dialyzed, and freeze-dried, yielding CRG (Σ-CRG). Then Σ-CRG was separated with 4% KCl into gelling (KCl-insoluble) and non-gelling (KCl-soluble) fractions, as described previously [30] and their structures were established according to the published protocols [30,33,34]. The monosaccharide composition of the polysaccharide, the sulfate ester content and molecular masses of CRGs were determined as [13,30]. FTIR spectra of the polysaccharides were recorded in films on Egunox 55 and Invenio S Fourier transform spectrophotometers (Bruker, Hamburg, Germany). The spectra were normalized by the monosaccharide ring skeleton absorption at 1074 cm^−1^ (120 scans with 4 cm^−1^ resolution). ^1^H and ^13^C nuclear magnetic resonance (NMR) spectra were recorded using a DRX-500 (125.75 MHz) spectrometer (Bruker, Hamburg, Germany) operating at 50 °C. The polysaccharides (3 mg) were deuterium-exchanged twice with heavy water (D_2_O, Chemical line, St. Petersburg, Russia) 0.6 mL by freeze-drying prior to examination in a solution of 99.95% D_2_O. Chemical shifts were described relative to the internal standard, acetone (δC 31.45, δH 2.25). The NMR data were acquired and processed using XWIN-NMR 1.2 software (Bruker).

### 4.3. Preparation of CRG/Ech Complex

The standardized substance Ech (pentahydroxyethylnaphthoquinone) in powder form (registration number in the Russian Federation is P N002362/01) [Russian State Register of Drugs (December 2016) Part 2] was obtained in G.B. Elyakov Pacific Institute of Bioorganic Chemistry FEB of the Russian Academy of Sciences, Vladivostok. A solution of 1% κ-CRG was prepared by dissolving 10 mg κ-CRG in 1 mL deionized water at 50 °C, while stirring on a stirrer. Stock solution of Ech in ethanol with a concentration of 10 mg/mL was prepared. To obtain a κ-CRG/Ech complex, 0.1 mL of a stock solution of Ech was added to 1 mL of a 1% κ-CRG solution. The mixture was left under stirring in a dark place at a temperature of 37 °C for 60 min. Thus, the solution of κ-CRG/Ech complex was obtained at a ratio of the initial components 10:1 (*w*/*w*). The concentration of Ech in the solution was measured by using absorption spectra at λ = 468 nm.

### 4.4. Molecular Docking

The structures of kappa-, beta-, and lambda-CRG tetrasaccharides were obtained using the molecular editor of the MOE 2020.0901 program [Molecular Operating Environment (MOE), 2020.09; Chemical Computing Group ULC, 1010 Sherbrooke St. West, Suite #910, Montreal, QC, Canada, H3A 2R7, 2020]. The CRG tetrasaccharides structures were solvated in the aqueous phase and optimized with the forcefield Amber10:EHT. The structure of the complex of the SARS-CoV-2 Delta variant RBD-ACE2 (PDB ID 7W9I) was used for the target proteins. The CRG binding sites for target proteins were determined using the Site Finder module of the MOE program. The calculations of the electrostatic potential of the molecular surface of target protein were carried out using the MOE program. Molecular docking of RBD SARS-CoV-2 and ACE2 with the CRG tetrasaccharides was performed using the Dock module of the MOE 2020.09 software. Default docking parameters were selected: the structures of 30 complexes were calculated with Score London dG, and the five most energetically advantageous complexes were optimized with Score GBVI/WSA dG. Contact analysis was carried out using the Ligand Interaction module of the MOE program.

### 4.5. Anti-SARS-CoV-2 Activity of the Tested Compounds

#### 4.5.1. Virus and Cell Culture

Virus SARS-CoV-2 strain SARS-CoV-2/human/RUS/367/2021 belonging to Delta genotype (Pango lineage: B.1.617.2; GISAID clade G; Nextstrain clade 21J) was obtained from the collection of the G.P. Somov Institute of Epidemiology and Microbiology of Rospotrebnadzor (GenBank ID ON692745.1). The strain was isolated from the autopsy material (human lung) of a patient with a clinically and laboratory-confirmed diagnosis of COVID-19 by sequential passages on a culture of African green monkey kidney epithelium cell culture (Vero E6) grown using Dulbecco’s Modified Eagle’s Medium (DMEM, Biolot, St. Petersburg, Russia), supplemented with 10% fetal bovine serum (FBS) (Biolot, St. Petersburg, Russia) and 100 U/mL of gentamycin (Dalkhimpharm, Khabarovsk, Russia) at 37 °C, 5% CO_2_. In the maintenance medium the FBS concentration decreased to 1%. Initial cell concentration in all experiments was 1 × 10^4^ cells/mL.

#### 4.5.2. Cytotoxicity of the Tested Compounds

For cytotoxicity and antiviral activity determination, the tested compounds—different types of CRGs, complex of CRG with Ech were diluted in DMEM. Remdesivir and ribavirin were used as reference compounds. Remdesivir^®^, freeze-dried powder for injections (Pharmasyntez, Irkutsk, Russia) was diluted in DMEM. Stock solutions (10 mg/mL) of Ech and Ribavirin^®^ (Sigma-Aldrich, St. Louis, MO, USA) were dissolved in dimethyl sulfoxide (DMSO, Sigma-Aldrich, St. Louis, MO, USA) and stored at -20 °C. For experiments, it was diluted with DMEM to a final concentration of 0.5% DMSO.

The cytotoxic effects of different types of CRGs, Ribavirin^®^ and Remdesivir^®^ in Vero E6 were assessed using the methylthiazolyltetrazolium bromide (MTT) test, as previously described [13,50]. Cytotoxicity was expressed as the 50% cytotoxic concentration (CC_50_) and maximum non-toxic concentration (MNTC) of the investigated compound causing 50% and 10% cell death compared to the control, respectively.

#### 4.5.3. Infectious Viral Titer of SARS-CoV-2

Virus titer was determined by the cytopathic effect (CPE) of the SARS-CoV-2 strain in Vero E6 cells using the MTT test [50]. Optical density was measured at 540 nm using an ELISA microplate reader (Labsystems Multiskan RC, Vantaa, Finland) with a reference absorbance at 620 nm. The cytopathogenicity level (*CPL*) of the strain was calculated using the formula:$$CPL=\left(1-\frac{D_{v}}{D_{0}}\right)\cdot 100\%,$$
where *D_v_* is the optical density of the infected sample, *D*_0_ is the optical density of the uninfected cell culture. The infectious titer of a virus was determined as the endpoint dilution of a virus stock at which *CPL* = 50% and was expressed as the 50% Tissue Culture Infectious Dose (TCID_50_) per mL. After the 8th passage on Vero E6 cells, the infectious titer of the SARS-CoV-2 strain was 5.8 lg TCID_50_/_mL_. All experiments involving infectious virus were performed in the biosafety level 3 facilities of the G.P. Somov Institute of Epidemiology and Microbiology Rospotrebnadzor.

#### 4.5.4. Anti-SARS-CoV-2 Activity of Different Types of CRGs

The inhibitory effects of the different types of CRGs on SARS-CoV-2-infection process in Vero E6 cells were evaluated by the cytopathic effect (CPE) inhibition assay using MTT-test and virus replication level (VRL) inhibition assay using real-time reverse transcription-polymerase chain reaction (real-time RT-PCR) assay. A monolayer of Vero E6 cells grown in 96-well plates was infected with 2.0 lg (TCID_50_/mL) SARS-CoV-2, and tested compounds were added at concentrations ranging from 10 to 250 μg/mL. The virus and compounds (1:1 *v*/*v*) were applied to Vero E6 cells monolayers simultaneously and incubated for one hour at 37 °C. After virus absorption, the virus–compound mixture was removed; the cells were washed, and maintenance medium was added. The plates were incubated for 5 days at 37 °C, 5% CO_2_.

#### 4.5.5. Anti-SARS-CoV-2 Activity of the κ-CRG/Ech Complex

To study the effect of the κ-CRG/Ech complex on the early stages of the SARS-CoV-2 life cycle, several schemes were studied, each of which was performed in three independent replicates using triplets of different concentrations of complex (from 10 to 250 μg/mL) as previously described [29]:

The pre-treatment of virus with κ-CRG/Ech complex. SARS-CoV-2 in dose of 2.0 lg (TCID_50_/mL) was mixed with different concentrations of studied complex (from 10 to 250 μg/mL) at a ratio 1:1 (*v*/*v*), incubated for one hour at 37 °C; then, the mixture was applied to the monolayer of cells. After 1 h adsorption at 37 °C, the cells were washed with PBS, overlaid by the maintenance medium, and followed by incubation for 5 days at 37 °C, 5% CO_2_.

The pretreatment of cells with κ-CRG/Ech complex. The monolayer of cells was pretreated with different concentrations of studied complex for 2 h at 37 °C. After washing, the cells were infected with 2.0 lg (TCID_50_/mL) of SARS-CoV-2 at 37 °C for one hour. Then, unabsorbed virus was removed by washing with PBS, and cells were incubated in the maintenance medium for 5 days at 37 °C, 5% CO_2_.

The simultaneous treatment of the cells with complex and virus. The monolayer of cells was infected with SARS-CoV-2 in dose 2.0 lg (TCID_50_/mL) and simultaneously treated by different concentrations of the complex (virus: compound, 1:1 *v*/*v*) for one hour at 37 °C. After virus adsorption, the mixture was removed; the cells were washed with PBS and incubated in maintenance medium for 5 days at 37 °C, 5% CO_2_.

The treatment of virus-infected cells with κ-CRG/Ech complex. The monolayer of cells was infected with 2.0 lg (TCID_50_/mL) SARS-CoV-2 at 37 °C for 1 h, and then the cells were washed with PBS and treated with different concentrations of the complex and incubated for 5 days at 37 °C, 5% CO_2_.

After the incubation, CPE inhibition and real-time RT-PCR assays were performed.

#### 4.5.6. CPE Inhibition Assay

The level of SARS-CoV-2 infection inhibition by the investigated compound was determined using the protection index *PI* calculated through the Pauwels’ formula [51]:$$PI=\frac{D_{ci}-D_{i}}{D_{0}-D_{i}}\cdot 100\%,$$
where $D_{0}$ is the optical density in the MTT-test for uninfected cells without compound, $D_{i}$—for the infected cells without compound, $D_{ci}$—for the infected cells in the presence of the compound.

The 50% inhibitory concentration (IC_50_) of each compound was determined as the concentration that reduced virus-mediated CPE by 50%. The selectivity index (SI) was calculated as the ratio of CC_50_ to IC_50_ for each compound.

#### 4.5.7. Virus Replication Level (VRL) Inhibition Assay

SARS-CoV-2 RNA was extracted from cell supernatants and amplified using “OM-Screen- 2019-nCoV-RT” RT-PCR kit (Sintol, Moscow, Russia) and Rotor-Gene Q (Qiagen, Hilden, Nordrhein-Westfalen, Germany) according to the manufacturer’s instructions. The PCR reaction was carried out under the following conditions: at 50 °C for 15 min; at 95 °C for 5 min; then 50 cycles: at 95 °C for 10 s, at 58 °C for 10 s, at 72 °C for 10 s. A negative sample was used as the amplification control for each run. The presence of viral RNA was assessed by the threshold cycle (*C_t_*) values; the *C_t_* ≥ 36 indicated the absence of the SARS-CoV-2 RNA in the samples. The effect of the test compound on the viral load was assessed using the inhibition coefficient (*IC*), which was calculated via the modified $2^{-∆C_{t}}$ method [39,52] developed by Professor M.Yu. Shchelkanov [53]:$$IC=\left(2^{\frac{C_{tci}-C_{ti}}{C_{t0}-C_{ti}}}-1\right)\cdot 100\%$$
where *C_t_*—the threshold PCR cycle number; *C_t_*_0_—threshold PCR cycle number for uninfected cells (cell control); *C_ti_*—threshold PCR cycle number for infected cells without the drug (virus control); *C_tci_*—threshold PCR cycle number for the infected cells treated by the investigated compound.

### 4.6. Statistical Analysis

The statistical analysis was performed using Statistica 10.0 software (StatSoft, Inc., Tulsa, OK, USA). ${CC}_{50}$ and ${IC}_{50}$ values were calculated using linear-logarithmic interpolation of the dose–response curve. The Kolmogorov–Smirnov test was used to assess the normality of the data distribution. Differences between groups were assessed by one-way ANOVA, followed by Bonferroni’s post hoc test. Results are presented as mean ± standard deviation (M ± SD) of three or more independent experiments and statistical significance was defined as *p* ≤ 0.05.

## 5. Conclusions

The antiviral activity of different types of carrageenans (kappa, lambda, kappa/beta-CRG) isolated from the red algae against SARS-CoV-2 was examined. The molecular docking was performed to determine potential interactions of CRGs with the receptor-binding domain (RBD) of SARS-CoV-2 and its cellular receptor—angiotensin-converting enzyme type 2 (ACE2). The highest binding energy of CRGs and ACE2 was observed for kappa-CRG that also exhibited the most significant antiviral effect. The ability of carrageenans, as well as the complex of kappa-CRG with Ech to inhibit SARS-CoV-2 replication in Vero E6 cells, was studied using the CPE inhibition and RT-PCR assays. These findings suggest carrageenans and κ-CRG/Ech complex could be promising anti-SARS-CoV-2 agents.

## Figures and Tables

**Figure 1 ijms-26-06175-f001:**
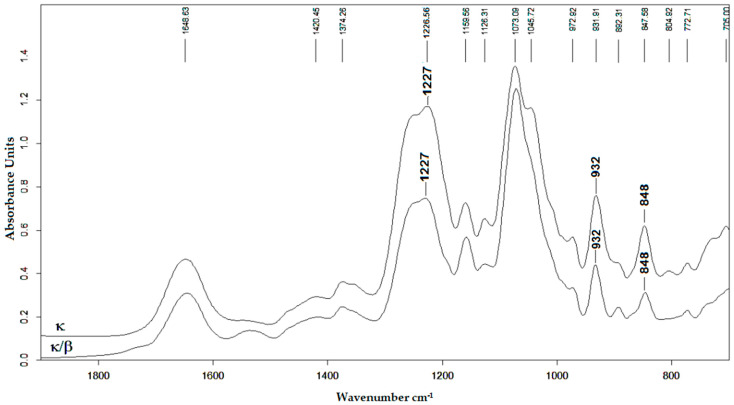
Infrared spectra of KCl—insoluble fraction of polysaccharides from *Chondrus armatus* (κ—top line) and *Tichocarpus crinitus* (κ/β—bottom line).

**Figure 2 ijms-26-06175-f002:**
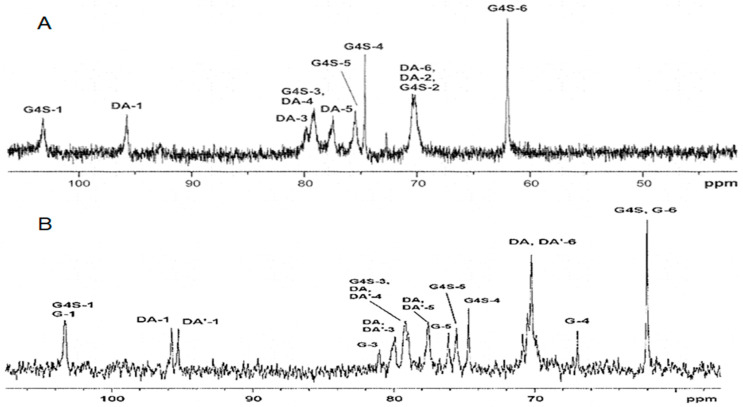
(**A**) ^13^C-NMR spectrum of KCI-insoluble fraction of polysaccharides from *Chondrus armatus*; (**B**) ^13^C-NMR spectrum of KCI-insoluble fraction from *T. crinitus*.

**Figure 3 ijms-26-06175-f003:**
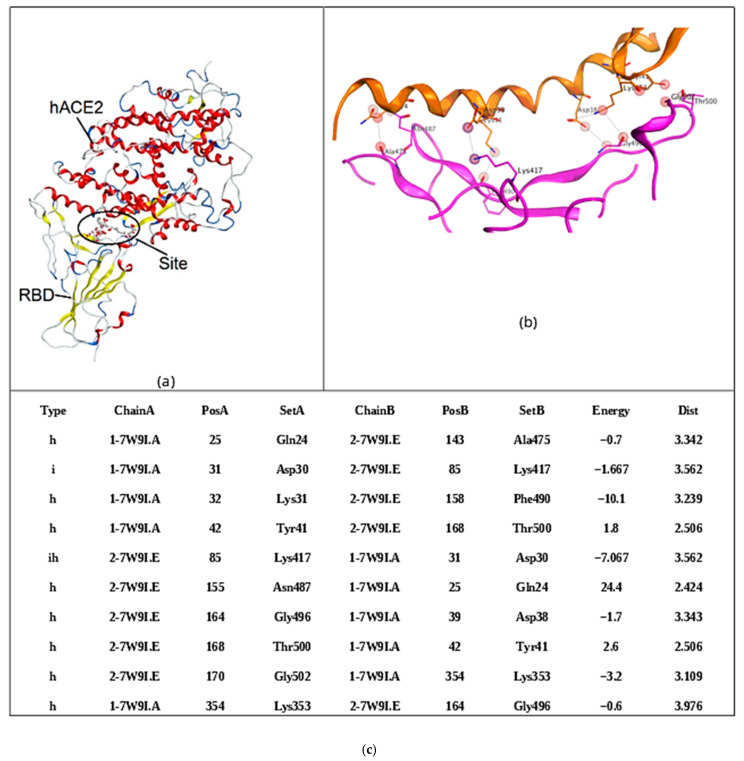
The SARS-CoV-2 Delta RBD and hACE2 binding site: (**a**) the structure of proteins is shown as ribbon, (**b**) the 3D structure of protein contacts. Structures are shown in orange (ACE2) and pink (RBD), (**c**) analysis of contacts between RBD Delta SARS-CoV-2 and the human ACE2 cell receptor (PBD ID 7w9i).

**Figure 4 ijms-26-06175-f004:**
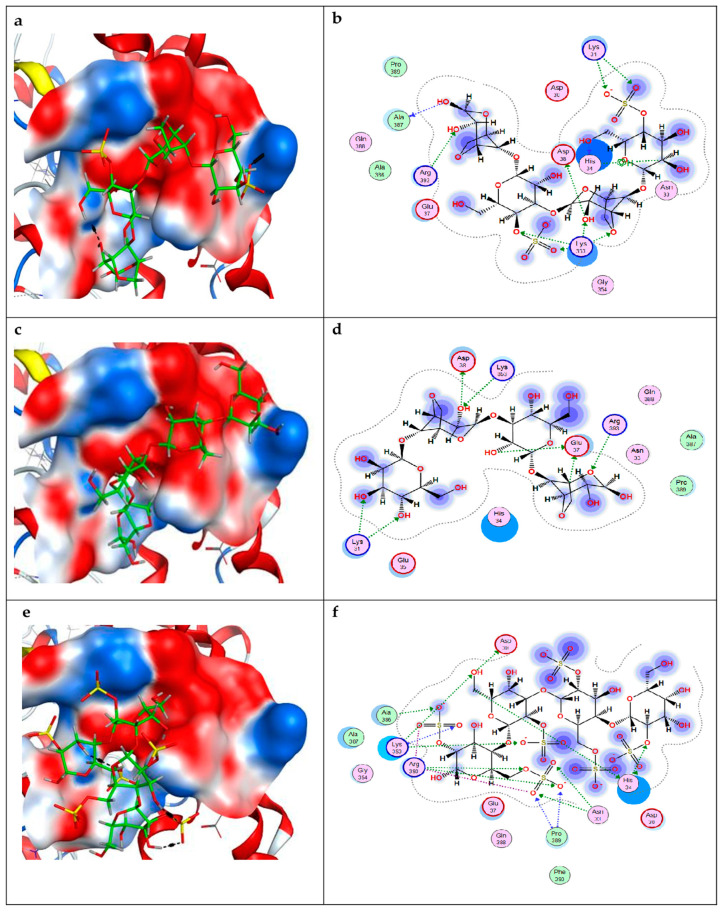
3D structures (**a**,**c**,**e**) and 2D diagrams (**b**,**d**,**f**) of contacts of κ-CRG (**a**,**b**), β-CRG (**c**,**d**) and λ-CRG (**e**,**f**) carrageenans complexes with human ACE2 (PdB ID 7w9i) at the RBD Delta binding site SARS-CoV-2. The structure of carrageenans is shown as stick in green. The electrostatic potential of the ACE2 molecular surface at the binding site is shown in red—electronegative, in blue—electropositive and in white—electroneutral.

**Figure 5 ijms-26-06175-f005:**
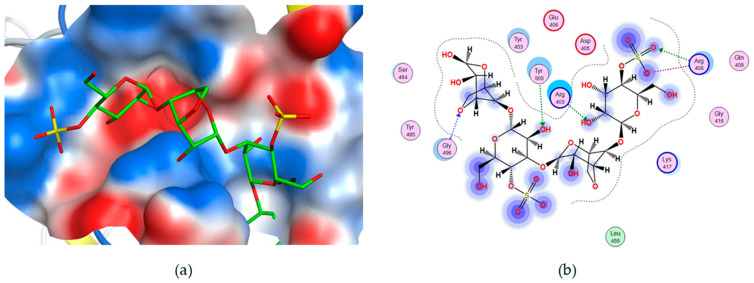
Three-dimensional structure (**a**) and 2D diagram (**b**) of contacts of κ-CRG complexes with the RBD binding site SARS-CoV-2 (PDB ID 7w9i). The structure of CRG is shown by the sticks in green. The electrostatic potential of the RBD molecular surface at the binding site is shown in red—electronegative, in blue—electropositive, and in white—electroneutral.

**Figure 6 ijms-26-06175-f006:**
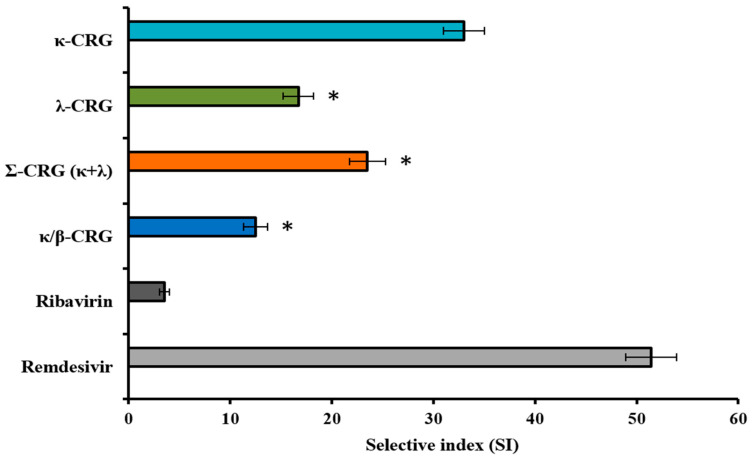
Anti-SARS-CoV-2 activity of different types of carrageenans. Vero E6 cells were infected with virus (2.0 lg TCID_50_/mL) and simultaneously treated with tested compounds. Selective index (SI) was calculated as the ratio of 50% cytotoxic concentration (CC_50_) to 50% inhibitory concentration (IC_50_) for each compound. * Significance of the differences between the parameters of κappa-CRG compared to CRG polysaccharides (λ-CRG, Σ-CRG (κ + λ), and κ/β-CRG) (*p* ≤ 0.05). Data represent mean ± SD from three independent experiments.

**Figure 7 ijms-26-06175-f007:**
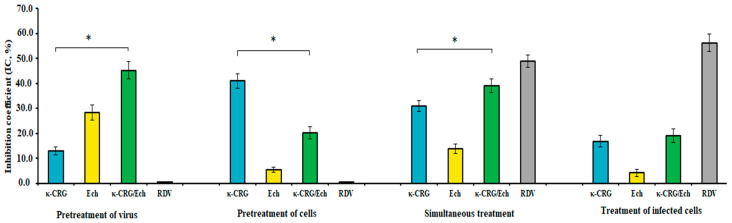
Anti-SARS-CoV-2 activity of the κ-CRG/Ech complex and its components (RT-PCR-assay). κ-CRG—κappa-carrageenan, Ech—echinochrome, κ-CRG/Ech—κ-CRG complex with echinochrome, RDV—remdesivir. Vero E6 cells were infected with virus (2.0 lg TCID_50_/mL) and treated with tested compounds in various schemes. The results of the RT-PCR assay were evaluated using the IC (%)—inhibition coefficient. * Significance of differences between the parameters of κ-CRG/Ech complex compared to its components (κappa-CRG and Ech) (*p* ≤ 0.05). Data represent mean ± SD from three independent experiments.

**Table 1 ijms-26-06175-t001:** The chemical structures of the CRGs.

Type of CRG	The Disaccharide Repeating Units	SO_3_Na% of Sample Weight	MW, kDa
Σ-CRG(κ + λ)kappa+lambda	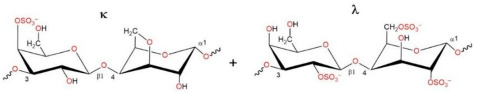	27.1	328
κ-CRGkappa-type	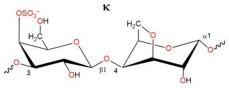	22.8	542
κ/βkappa/beta-CRG	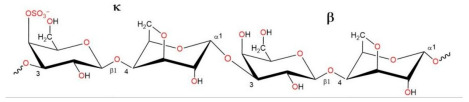	18.7	229
λ-CRGlambda–type	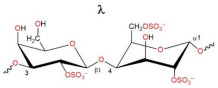	30.9	145

**Table 2 ijms-26-06175-t002:** Anti-SARS-CoV-2 action of the κ-CRG/Ech complex and its components (CPE inhibition assay).

Compounds	CC_50_ (µg/mL)	Pretreatment of Virus		Pretreatment of Cells		Simultaneous Treatment		Treatment of Infected Cells	
		IC_50_ (µg/mL)	SI	IC_50_ (µg/mL)	SI	IC_50_ (µg/mL)	SI	IC_50_ (µg/mL)	SI
κ-CRG	>2000	75 ± 9	27 ± 3 *	28 ± 3	71 ± 9 *	62 ± 6	32 ± 3 *	90 ± 9	22 ± 3
Ech	142 ± 6	4 ± 0.5	35 ± 4 *	89 ± 9	1.6 ± 0.2 *	41 ± 5	3.5 ± 0.4 *	92 ± 9	1.5 ± 0.2 *
κ-CRG/Ech	>1000	13 ± 2	77 ± 8	49 ± 5	20 ± 2	24 ± 3	42 ± 4	43 ± 5	23 ± 3
Ribavirin	730 ± 80	NA	NA	NA	NA	207 ± 25	3.5 ± 0.4	158 ± 17	4.6 ± 0.5
Remdesivir	72 ± 9	NA	NA	NA	NA	1.4 ± 0.1	51 ± 6	0.6 ± 0.1	120 ± 13

Note: κ-CRG—κappa-carrageenan, Ech—echinochrome, κ-CRG/Ech—κ-CRG complex with echinochrome; IC_50_—50% inhibiting concentration, SI—selective index, NA—not active. Ribavirin and remdesivir were used as reference compounds. Data are expressed as means ± SD of three independent experiments. * Significance of differences between the parameters of κ-CRG/Ech complex compared to its components (κappa-CRG and Ech) (*p* ≤ 0.05).

## Data Availability

The original data are available from the correspondent author on request.

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
