# Peer review of "Carrageenans and the Carrageenan-Echinochrome Complex as Anti-SARS-CoV-2 Agents"

_ijms, 2025, doi:10.3390/ijms26136175_

Round 1
Reviewer 1 Report
Comments and Suggestions for Authors
This study comprehensively evaluated the anti-SARS-CoV-2 activity of different structural types of carrageenans (κ-, λ-, κ/β-CRG) and their complex with echinochrome (Ech) (κ-CRG/Ech), combining molecular docking with in vitro cellular experiments (CPE inhibition and RT-PCR). The study design is sound, and the data are detailed, providing valuable experimental evidence for the application of naturally sourced polysaccharides against SARS-CoV-2. However, the manuscript requires further refinement in language expression, methodological details, rigor of results interpretation, and reference formatting. It is recommended for acceptance after revision. For specific issues, please refer to the contents in the attachment.

Author Response
Response to Reviewer 1
We are grateful to the reviewers for the deep analysis of our manuscript and useful remarks. Please find below our response to the comments of Reviewer 1:
- Reviewer’s comments
Abstract: Change "can to act" to "can act"; Change "reduction in viral RNA levels with inhibition coefficient (IC) 45%" to "reduced viral RNA levels by 45% (IC=45%)".
Our response:
Corrected
- Reviewer’s comments
Introduction: Change "that have helped control the pandemic" (Page 1) to "which have helped control".
Our response:
Corrected
- Reviewer’s comments
Results 2.2: Change "us used tetrasaccharides" to "we used tetrasaccharides"; Change "docking score ... changes in the series" to "docking scores ... ranked in the order".
Our response:
Corrected
- Reviewer’s comments
Discussion: Change "This result in the formation" to "This results in the formation".
Our response:
Corrected
- Reviewer’s comments
Conclusions: The phrase "complex action" in "The complex action of different types of carrageenans... was examined" is ambiguous. Suggest changing to "combined antiviral activity".
Our response:
Corrected
- Reviewer’s comments
"echinochrome" should be used consistently throughout the manuscript as either "echinochrome (Ech)" or "Ech".
Our response:
Corrected
(7) Reviewer’s comments
Ensure consistent designation of the "κ-CRG/Ech complex" (avoid sometimes using "CRG/Ech").
Our response:
Corrected
(8) Reviewer’s comments
Professional English editing is recommended to correct grammatical errors and simplify complex sentence structures.
Our response:
The manuscript was proofread by a native English speaker, certificate English proofreading included.
(9) Reviewer’s comments
Figure 6: The asterisk (*) denoting significant differences (p≤0.05) does not specify the comparison groups (e.g., κ-CRG vs. other CRGs). The comparison groups must be clearly indicated in the figure legend or directly on the figure.
Our response:
Figure 6 corrected: the asterisk (*) indicates comparison groups (κ-CRG compared to other CRGs) both in the figure and in the figure legend.
(10) Reviewer’s comments
Table 2: The footnote states the asterisk (*) denotes significant differences for κ-CRG/Ech vs. κ-CRG, but lacks comparison data involving the Ech group itself. Data comparing Ech to relevant groups should be included.
Our response:
Table 2 corrected: asterisk (*) indicates significant differences between parameters of the κ-CRG/Ech complex and its components (compared to both κappa-CRG and Ech).
(11) Reviewer’s comments
Figures 1 & 2: Key absorption peaks or chemical shift assignments are not labeled. Detailed annotations identifying critical features are required.
Our response:
We have designated the main adsorption bands in Fig. 1 and have corrected the values of these bands in the text. We have placed new Fig 1 in the text of the article.
On Figure 2 the main signals in the anomeric region, characterizing the corresponding disaccharide units, are indicated in the figures themselves (for example, the signal at 103.2 ppm corresponds to G4S).
(12) Reviewer’s comments
Methods (Titer Determination): Details of the TCID50 assay are insufficient. Provide a more detailed description of the method.
Our response:
We have described the determination of the virus titer using the TCID50 assay in more detail and placed it in the text of the article in the Materials and Methods section, paragraph 4.5.3.
(13) Reviewer’s comments
Methods (Complex Preparation): The rationale for selecting the 10:1 ratio of kappa-CRG to Ech is not provided. Data from concentration optimization experiments should be included to justify this ratio.
Our response:
We used this ratio of polysaccharide to echinochrome to obtain a soluble complex that has previously shown high antiherpetic activity.
(14) Reviewer’s comments
Statistics: The use of the "Wilcoxon test for related samples" for multiple group comparisons (e.g., Figure 6) may be inappropriate. It is strongly recommended to use ANOVA followed by a post-hoc test (e.g., Tukey).
Our response:
Statistical analysis was corrected: the Kolmogorov-Smirnov test was used to assess the normality of data distribution. Differences between groups were assessed using one-way analysis of variance (ANOVA) followed by Bonferroni post hoc test.
(15) Reviewer’s comments
Ensure all Figures (1-7) have image resolution ≥ 300 dpi and complete, descriptive legends.
Our response:
Corrected: All figures (1-7) have an image resolution of ≥ 400 dpi.
(16) Reviewer’s comments
Ref. 4: Volume and issue numbers are missing. Ref. 22: Inconsistent use of "et al." in the author list. Ensure consistency with journal style.
Our response:
Corrected
(17) Reviewer’s comments
Ref. 3: Journal abbreviation "JRHS" should be written in full as "Journal of Research in Health Sciences" (or as per the journal's required abbreviation style).
Our response:
Corrected
Reviewer 2 Report
Comments and Suggestions for Authors
Review Comments on the Study: Antiviral Potential of Carrageenans Against SARS-CoV-2
Overview: This study investigates the antiviral activity of different types of carrageenans (CRGs)—kappa, lambda, and kappa/beta—derived from red algae of the Pacific coast, and their complex with echinochrome (Ech) against SARS-CoV-2. The study utilizes both computational molecular docking and experimental biological assays to explore their mechanisms of action and efficacy.
1. Scientific Clarity & Organization
-
Strength: The study addresses a highly relevant issue—the discovery of natural marine-derived antiviral agents against SARS-CoV-2. The dual approach of molecular docking and biological assays enhances the scientific robustness.
-
Comment: The abstract would benefit from a more structured presentation. Organizing the findings by method (e.g., computational studies first, followed by in vitro results) would improve readability.
2. Scientific Accuracy & Depth
-
Docking Studies:
-
The mention of hydrogen bonding and ionic interactions is scientifically accurate.
-
Suggestion: Include details about specific interacting residues on ACE2 or RBD to strengthen mechanistic interpretation.
-
-
Antiviral Assays:
-
The use of cytopathic effect (CPE) inhibition and RT-PCR is methodologically sound.
-
Suggestion: Include IC50 or EC50 values for better clarity.
-
-
Selectivity Index (SI):
-
Highlighting the SI value for kappa-CRG (33) and CRG/Ech complex (>70) is impactful.
-
Suggestion: Provide the cytotoxic concentration (CC50) values used to calculate SI.
-
3. Language & Grammar Improvements Several grammatical and typographical issues should be corrected for better clarity:
-
"can to act" → "can act"
-
"re-duction" → "reduction"
-
"interaction of CRGs with ACE2 and RBD occurred due to..." → "CRGs interacted with ACE2 and RBD via hydrogen bonding and ionic interactions."
-
"highest binding energy" → consider rephrasing to "strongest binding affinity," and clarify if more negative energy values indicate higher affinity.
4. Scientific Impact
-
The CRG/Ech complex’s SI (>70) is promising and suggests enhanced efficacy.
-
Suggestion: Elaborate on the possible synergistic mechanisms by which Ech improves CRG activity (e.g., increased viral binding affinity, cell membrane stabilization, or enhanced uptake).
5. Suggestions for Improvement
-
Clarify Experimental Methods: Briefly mention the docking software and the assay conditions used (e.g., multiplicity of infection, treatment times).
-
Compare with Standard Antivirals: Position your findings relative to standard treatments like Remdesivir or Paxlovid.
-
Add Mechanistic Hypotheses: Discuss how CRGs may block viral entry—possibly by mimicking heparan sulfate or interfering with RBD-ACE2 interactions.
Conclusion: This paper provides compelling evidence for the antiviral potential of carrageenans, especially kappa-CRG and its complex with echinochrome. With improved organization, inclusion of quantitative data, and expanded mechanistic discussion, this work can make a significant contribution to the field of natural antiviral therapeutics.
Author Response
Response to Reviewer 2
We are grateful to the reviewers for the deep analysis of our manuscript and useful remarks. Please find below our response to the comments of Reviewer 2:
- Reviewer’s comments
The abstract would benefit from a more structured presentation. Organizing the findings by method (e.g., computational studies first, followed by in vitro results) would improve readability.
Our response:
In accordance with your advice, we have slightly structured the abstract.
The diversity of structural types of carrageenans (CRGs) - sulfated polysaccharides of red algae, - determines their different biological activities. The different types of CRGs (kappa, lambda, kappa/beta-CRGs) were isolated from the red algae of the Pacific coast. Molecular docking was performed to determine potential interactions of CRGs with the receptor-binding domain (RBD) of SARS-CoV-2 and its cellular receptor - angiotensin-converting enzyme type 2 (ACE2). CRGs interacted with ACE2 and RBD via hydrogen bonding and ionic interactions. The strongest binding affinity of CRGs and ACE2 was observed for kappa-CRG. Molecular docking was confirmed by results studying the effects of CRGs against SARS-CoV-2 in vitro. The ability of CRGs, as well as the complex CRG with sea urchin echinochrome (Ech) to inhibit SARS-CoV-2 replication in Vero E6 cells was studied using cytopathic effect (CPE) inhibition and RT-PCR assays. The simultaneous treatment of cells with CRGs and the virus revealed that kappa-CRG exhibited the most significant antiviral effect among all the polysaccharides, with a selective index (SI) of 33. The kappa-CRG/Ech complex exhibited the highest virucidal effect on SARS-CoV-2 particles with an SI above 70 (more than 2 times higher than that of CRG and Ech) and reduced viral RNA levels by 45% (IC=45%). Our results illustrate that CRGs and kappa-CRG/Ech complex can to act as protective agents against SARS-CoV-2.
- Reviewer’s comments
Include details about specific interacting residues on ACE2 or RBD to strengthen mechanistic interpretation.
Our response:
Details of the specific interacting residues on ACE2 or RBD are clearly presented in the Tables S1 and Tables S2 as well as the binding energy calculations. Due to the article being overloaded with figures and tables, we have placed these tables in the Supplementary materials.
According to your suggestion, we have made a more correct reference to these tables in the text.
Change Page 271-276
Analysis of contacts between CRGs and ACE2 showed the involvement of lysine residues in the binding of all types of CRGs (Figure 4; Table S1). The contacts of CRGs tetrasaccharides with the human ACE2 domain are presented in Table S1 (Supplementary material). According to the calculated data given in Table S1, the interaction of CRGs with ACE2 occurs due to the formation of H-donor, H-acceptor and ionic bonds. As can be seen the highest binding energy of CRGs and ACE2 was observed for kappa-CRG (Table S1).
Change Page 291-293
As in the case of ACE 2, the interaction of CRGs tetrasaccharides with RBD occurs due to the formation of H-donor, H-acceptor and ionic bonds, the energy calculation of which is presented in Table S2.
- Reviewer’s comments
Include IC50 or EC50 values for better clarity. Provide the cytotoxic concentration (CC50) values used to calculate SI.
Our response:
Details of the inhibitory effects of the different types of CRGs on SARS-CoV-2 infection process in Vero E6 cells (СС50, IC50) presented in the Tables S3 in the Supplementary materials.
When studying the cytotoxicity of the compounds, we showed that the studied carrageenans have low cytotoxicity: their CC50 is more than 2000 μg/ml.
It was found that the all investigated carrageenans had antiviral activity, which was confirmed by the inhibition of the cytopathic effects induced by SARS‐CoV‐2 (Figure 6 and Table S3). The kappa-CRG demonstrated the greatest antiviral effect: its IC50 = 61 µg/mL and SI, indicating the efficacy and safety of the compound, was 33, while the values of these parameters for less sulfated kappa/beta-CRG were 2.5 times lower (IC50 = 160 µg/mL and SI 12.5) than for kappa-CRG. Meanwhile, ribavirin showed the lowest anti-SARS-CoV-2 activity (IC50 = 207 µg/mL, SI 3.5), and remdesivir - the highest (IC50 = 1.4 µg/mL, SI 51).
- Reviewer’s comments
Several grammatical and typographical issues should be corrected for better clarity.
Our response:
The manuscript was proofread by a native English speaker, certificate English proofreading included.
- Reviewer’s comments
Elaborate on the possible synergistic mechanisms by which Ech improves CRG activity (e.g., increased viral binding affinity, cell membrane stabilization, or enhanced uptake).
Our response:
We have previously shown on the herpes virus that Ech exhibits a virucidal effect, while CRG mainly blocks the attachment of the virus to the cell [29]. Their combined synergistic effect is also manifested in an increase in the antiviral activity of the complex against SARS -2.
- Reviewer’s comments
Clarify Experimental Methods: Briefly mention the docking software and the assay conditions used (e.g., multiplicity of infection, treatment times).
Our response:
The CRG binding sites for target proteins were determined using the Site Finder module of the MOE program. The calculations of the electrostatic potential of the molecular surface of target protein were carried out using the MOE program. Molecular docking of RBD SARS-CoV-2 and ACE2 with the CRG tetrasaccharides was performed using the Dock module of the MOE 2020.09 software.
To study the anti-SARS-CoV-2 activity of different types of CRGs and the κappa-CRG/Ech complex, an infectious dose of the virus of 2 lg TCID50/mL was used. The virus and compounds (1:1 v/v) were applied to Vero E6 cells monolayers simultaneously and incubated for one hour at 37 ºC. After virus absorption, the virus–compound mixture was removed; the cells were washed, and maintenance medium was added. The plates were incubated for 5 days at 37 °C, 5% CO2.
- Reviewer’s comments
Compare with Standard Antivirals: Position your findings relative to standard treatments like Remdesivir or Paxlovid.
Our response:
Remdesivir and Ribavirin were used as reference drugs in the study of the anti-SARS-COV-2 activity of various types of carrageenans (Table 2, Figures 6 and 7, and in the supplementary material - Table S3 and S4).
- Reviewer’s comments
Add Mechanistic Hypotheses: Discuss how CRGs may block viral entry—possibly by mimicking heparan sulfate or interfering with RBD-ACE2 interactions.
Our response:
In accordance with your advice, we included this text in the Discussion section
The in silico results allowed us to better understand the complex mechanism underlying the action of CRG carrageenan and its complex against SARS-CoV-2 in vitro. AS it know SARS-CoV-2 uses its receptor-binding domain RBD (Spike glycoprotein) to interact and gain access to host cells by binding to the ACE2 receptor [20,21]. The S protein-ACE2 interaction is the primary key to virus entry. In our case, molecular docking clearly showed that CRG binds to both RBD and ACE2 and thus blocking their interaction and penetration of the virus into cells. At the same time, CRG exhibits a high affinity for ACE and, thus, already at the initial stage of infection, competes with RBD and suppresses the attachment of the virus to the cell. Thus, we believe that different mechanisms are involved in the antiviral action of the studied polysaccharides, the combination of which increases the effectiveness of the drugs.